# Inadequate Weight Gain According to the Institute of Medicine 2009 Guidelines in Women with Gestational Diabetes: Frequency, Clinical Predictors, and the Association with Pregnancy Outcomes

**DOI:** 10.3390/jcm9103343

**Published:** 2020-10-18

**Authors:** Xinglei Xie, Jiaming Liu, Isabel Pujol, Alicia López, María José Martínez, Apolonia García-Patterson, Juan M. Adelantado, Gemma Ginovart, Rosa Corcoy

**Affiliations:** 1Departament de Medicina, Universitat Autònoma de Barcelona, Bellaterra, 08193 Barcelona, Spain; xxienglei@santpau.cat (X.X.); ljiaming@santpau.cat (J.L.); 2Servei d’Endocrinologia i Nutrició, Hospital de la Santa Creu i Sant Pau, 08041 Barcelona, Spain; ipujol@santpau.cat (I.P.); alopezar@santpau.cat (A.L.); mmartinezr@santpau.cat (M.J.M.); 3Institut de Recerca, Hospital de la Santa Creu i Sant Pau, 08041 Barcelona, Spain; 31178agp@comb.cat; 4Servei de Ginecologia i Obstetricia, Hospital de la Santa Creu i Sant Pau, 08041 Barcelona, Spain; adelantadojm@gmail.com; 5Servei de Pediatria, Hospital de la Santa Creu i Sant Pau, 08041 Barcelona, Spain; gginovart.germanstrias@gencat.cat; 6CIBER-BBN, 28029 Madrid, Spain

**Keywords:** gestational diabetes mellitus, Institute of Medicine, weight gain, length of follow-up, pregnancy outcome

## Abstract

Background: In the care of women with gestational diabetes mellitus (GDM), more attention is put on glycemic control than in factors such as gestational weight gain (GWG). We aimed to evaluate the rate of inadequate GWG in women with GDM, its clinical predictors and the association with pregnancy outcomes. Methods: Cohort retrospective analysis. Outcome variables: GWG according to Institute of Medicine 2009 and 18 pregnancy outcomes. Clinical characteristics were considered both as GWG predictors and as covariates in outcome prediction. Statistics: descriptive, multinomial and logistic regression. Results: We assessed 2842 women diagnosed with GDM in the 1985–2011 period. GWG was insufficient (iGWG) in 50.3%, adequate in 31.6% and excessive (eGWG) in 18.1%; length of follow-up for GDM was positively associated with iGWG. Overall pregnancy outcomes were satisfactory. GWG was associated with pregnancy-induced hypertension, preeclampsia, cesarean delivery and birthweight-related outcomes. Essentially, the direction of the association was towards a higher risk with eGWG and lower risk with iGWG (i.e., with Cesarean delivery and excessive growth). Conclusions: In this cohort of women with GDM, inadequate GWG was very common at the expense of iGWG. The associations with pregnancy outcomes were mainly towards a higher risk with eGWG and lower risk with iGWG.

## 1. Introduction

Gestational diabetes mellitus (GDM) entails risks for the mother and the newborn, both at short and long-term [1,2]. Among others, it increases the risk of preeclampsia, caesarean section, and future type 2 diabetes for the mother [3,4] and for the baby it increases the risk of diabetic fetopathy at short term and metabolic syndrome at long term [5]. In turn, high pre-pregnancy body mass index (BMI) is associated with higher risks of gestational hypertension and GDM [6] as well as with unfavorable pregnancy outcomes [7,8]. Likewise, gestational weight gain (GWG) is related to adverse pregnancy outcomes. In the general obstetric population, excessive GWG (eGWG) is associated with a higher risk of hypertensive disorders of pregnancy, GDM, caesarean section, postpartum weight retention, large for gestational age (LGA) and macrosomic newborns [9,10]. In turn, insufficient GWG (iGWG) is associated with a higher risk of preterm birth and small for gestational babies (SGA) [11,12,13]. In 2009, the Institute of Medicine (IOM) provided specific guidelines regarding the recommended GWG according to BMI categories to optimize pregnancy outcome [14].

In diabetic pregnancy, inadequate GWG is prevalent both in women with GDM and those with pregestational diabetes [13,15,16] and has also been related to perinatal outcome. In a meta-analysis on GWG and pregnancy outcomes in women with GDM, eGWG was associated with a higher frequency of pregnancy-induced hypertension (PIH), cesarean delivery, LGA newborns and macrosomia in relation to an adequate GWG (aGWG) whereas iGWG was associated with a lower risk of LGA and macrosomia. The risk of LGA was ≃double with eGWG and ≃0.70 with iGWG [13]. Articles addressing this point in our background are limited [17,18,19] and, therefore, we aimed to evaluate the frequency of inadequate GWG in women with GDM attended in our center, its clinical predictors and the association with pregnancy outcomes.

## 2. Materials and Methods 

### 2.1. Study Design

The present study is a retrospective analysis of data collected prospectively in the database of the Endocrinology and Pregnancy Clinic at Hospital de la Santa Creu i Sant Pau. All patients provided written informed consent for inclusion in the database and the study has been approved by the Ethics Committee of the Institut de Recerca de l’Hospital de la Santa Creu i Sant Pau and has been performed in accordance with the Declaration of Helsinki. We evaluated patients with GDM who were attended in our center between 1 January 1985 and 31 December 2011. Throughout this period, GDM has been diagnosed after National Diabetes Data Group Criteria, continued after an evaluation of the potential impact of Carpenter and Coustan criteria [20,21]. Treatment approach has essentially kept constant: normocaloric diet (usually ranging from 1800 to 2200 calories per day), self-monitoring of ketonuria (aiming at its absence) and of blood glucose (aiming at fasting/preprandial <90 mg/dL and 1 h postprandial <120 mg/dL unless fetal growth is <10th centile). 

### 2.2. Variables Collected

Adequacy of GWG was categorized according to IOM 2009 into insufficient, adequate (reference category) and excessive weight gain. GWG was calculated as the difference between the last weight available during pregnancy and prepregnancy (see Appendix A
Table A1). As potential independent variables, we have considered maternal ethnicity (non-Caucasian), age at the beginning of pregnancy, maternal anthropometry (height and prepregnancy BMI category), family history of diabetes, prior history of abnormal glucose tolerance (GDM/impaired fasting glucose/glucose intolerance), prior pregnancy, unfavorable obstetric history (macrosomia, pregnancy-induced hypertension, recurrent miscarriage, non-syndromic malformation, unexplained fetal death, polyhydramnios, pyelonephritis), multiple pregnancy, smoking habit during pregnancy (non-smoker at the beginning of pregnancy, quitter or active smoker during pregnancy), characteristics of GDM diagnosis (gestational age at diagnosis, season, glycemic values, number of abnormal glucose values, autoimmunity against beta cell), gestational age at delivery and length of specific follow-up at the Diabetes and Pregnancy Clinic. 

We addressed 4 maternal and 14 neonatal outcomes, defined as follows: PIH (blood pressure ≥ 140/90 mmHg, ×2 times separated ≥ 6 h, starting at a gestational age ≥ 20 weeks or worsening chronic hypertension), preeclampsia (PIH, accompanied by proteinuria), insulin treatment, cesarean delivery (total), fetal scalp blood pH < 7.25 [22], preterm birth (defined as a gestational age at birth less than 37 complete weeks), Apgar 5 min < 7 [23], arterial pH < 7.10 [24], significant obstetric trauma, LGA newborn (birth weight >90% centile for the same gestational age and sex [25], macrosomia (defined as a birth weight ≥ 4000 g), small for gestational age (SGA) newborn (birth weight <10% centile for the same gestational age and sex) [25], neonatal hypoglycemia (Cornblath criteria applied to capillary blood) [26], neonatal jaundice requiring treatment [27], neonatal respiratory requiring treatment distress [28], neonatal hypocalcemia [29], polycythemia [30] and perinatal mortality (intrauterine or until 28 days postpartum taking into account fetal viability: before 1991, <28 completed weeks; 1991–1994, <26 completed weeks; 1995–1999, <24 completed weeks and from 2000 onwards, <23 completed weeks). As potential independent variables for pregnancy outcomes in addition to GWG, we considered the following characteristics: maternal ethnicity, age at the beginning of pregnancy, maternal anthropometry (height and pre-pregnancy BMI category), prior pregnancy, multiple pregnancy, smoking habit during pregnancy (non-smoker at the beginning of pregnancy, quitter or active smoker during pregnancy), characteristics of GDM diagnosis (gestational age at diagnosis, glycemic values), delay between diagnosis and initiation of specific follow-up, first HbA1c after diagnosis, average HbA1c in the third trimester) and fetal sex. In multiple pregnancies, a variable of concordant fetal sex was computed to be used in the analysis of fetal outcomes; for maternal outcomes, the sex of the fetus with higher risk was used. The variables above were used for the adjusted analysis of all outcomes with the exception of average HbA1c in the third trimester that was excluded for the adjusted analysis of insulin treatment. Over the years, different methods have been used to determine glycated hemoglobin. Currently, it is measured in whole blood using cation exchange HPLC (Variant II Turbo HbA1c, Bio-Rad Laboratories, Hercules, CA, USA). The results obtained with the different methods have been collected as SD around the mean and translated into values referred to DCCT.

### 2.3. Statistical Analysis

Statistical analyses were performed using the SPSS version 26.0 software package. Descriptive results are expressed as mean and standard deviation (SD) or P50 (P25–P75) for continuous variables according to their normal or non-normal distribution. Categorical variables are expressed as percentages. We compared characteristics between the GWG categories using a Chi-square test or a Kruskal–Wallis test as appropriate (the variable being categorical or quantitative, non-normally distributed). Imputations were not used to deal with missing data.

To address the clinical characteristics associated with GWG according to IOM, we performed a multivariate multinomial logistic regression analysis with a forward method using as dependent variable weight gain according to IOM (reference category: aGWG) and using as potential predictors the variables with a *p* value <0.100 in the bivariate analysis. 

To determine the association of GWG according to IOM with pregnancy outcomes, we performed a logistic regression analysis (forward method) using aGWG as the reference category. Results were expressed as non-adjusted odds ratios (OR) and adjusted odds ratios (aOR) and 95% confidence intervals (95% CI). For the adjusted analysis, we fed in the model all the potential predictors indicated above.

A *p* value <0.05 was used as the cut-off for significance in the multivariate analysis. All *p* values were two-sided.

## 3. Results 

A total of 2842 pregnant women with GDM were attended during the period and information regarding GWG according to IOM and potential predictors was available in 2700 (2594 with singleton pregnancies, 106 with multiple pregnancies, 95.0% of the target population), with analyses being performed in this group. A total of 2818 babies were born from these mothers (2594 from singleton, 224 from multiple pregnancies). Figure 1 displays the flowchart of patient inclusion. Results on preeclampsia are limited to the last period when information on proteinuria was included in the database (N = 377).

Table 1 shows the characteristics of these women. Main characteristics are as follows: most women had a normal prepregnancy BMI, 62.3% women had been pregnant before, GDM was diagnosed at a gestational age of 29 weeks and length of specific follow-up was 7 weeks. Median GWG was 10.2 kg. The distribution of GWG according to IOM was: 50.3% insufficient, 31.6% adequate and 18.1% excessive (Figure 2). In the bivariate analysis, 13 out of the 20 characteristics considered as potential independent variables had a *p* value <0.100 among the three groups of women. Among these variables, prepregnancy overweight/obesity and smoking habit at the beginning of pregnancy (women quitting smoking/active smokers during pregnancy) increased throughout the categories of GWG. The length of follow-up for GDM was negatively associated with GWG (Table 1, Appendix B, Figure A1).

Results of the multinomial logistic regression are presented in Table 2 with seven characteristics being significantly associated with categories of GWG after IOM: non-Caucasian ethnicity, height, prepregnancy BMI category, unfavorable obstetric history, smoking habit, gestational age at delivery and length of follow-up. 

For iGWG, independent variables were maternal height, prepregnancy overweight/obesity, unfavourable obstetric history, smoking habit (active smoker and quitter during pregnancy), gestational age at delivery and length of follow-up, all of them negatively associated with the exception of length of follow-up (OR 1.035 per week; 95% CI 1.018–1.052). For eGWG, independent variables were non-Caucasian ethnicity, prepregnancy overweight/obesityand smoking habit (active smoker and quitter during pregnancy), all of them positively associated with eGWG. The length of follow-up was negatively associated with borderline significance. 

According to the recommendations of Institute of Medicine 2009, approximately 50% of women with gestational diabetes had insufficient weight gain during pregnancy, 18% excessive weight gain and 32% adequate weight gain.

Pregnancy outcomes according to IOM 2009 are presented in Table 3, with significant associations with three maternal (PIH, preeclampsia, cesarean delivery) and eight fetal outcomes (preterm birth, arterial pH < 7.1, LGA, macrosomia, SGA, jaundice requiring treatment, respiratory distress requiring treatment and neonatal hypocalcemia).

Unadjusted and adjusted OR resulting from logistic regression are presented in Table 4. In the adjusted analysis, GWG according to IOM was significantly associated with PIH, preeclampsia, cesarean delivery, LGA, macrosomia and SGA. With the exception of SGA, eGWG was associated with higher risks and iGWG with lower risks even when for some variables (PIH, preeclampsia and SGA), significance was present for GWG according to IOM but not for individual categories of iGWG and eGWG. Additionally, iGWG was associated with neonatal hypocalcemia (aOR 4.557, 95% CI 1.037–20.003) even when the global association of GWG did not reach significance (overall *p* 0.133).

In addition to weight gain itself, women in the different categories of gestational weight gain according to IOM differed in 13 out of 20 characteristics.

## 4. Discussion

The present study in a large cohort of women with GDM and satisfactory overall pregnancy outcomes has shown that GWG was very frequently outside IOM recommendations and GWG, in turn, was associated with PIH, preeclampsia, cesarean delivery, LGA, macrosomia and SGA with eGWG essentially linked with higher risks and iGWG with lower ones.

### 4.1. Prevalence and of Inadequate GWG

The distribution of GWG according to IOM in this cohort (50.3% insufficient, 31.6% adequate, 18.1% excessive) is far away from IOM recommendations. The rate of eGWG is similar to that in the study of Barquiel et al., also in Spain (14.7%) [19]—iGWG was not reported in this article [19]. In a meta-analysis of more than 80,000 patients with GDM, rates of inadequate GWG were also very high (30% insufficient, 34% adequate and 37% excessive) [13] and this was also the case in a recent article on this subject [31]. However, the GWG distribution in this study is clearly shifted towards iGWG whereas in other publications on this subject, the percentage of insufficient and excessive GWG is more balanced. We partially attribute this difference to the fact that, in the center, the glycemic control goals (90 mg/dL basal/preprandial, <120 mg/dL 1 h postprandial) are tighter than usual (95 mg/dL basal/preprandial, <140 mg/dL 1 h postprandial). This may facilitate that in order to avoid or delay insulin treatment, there is a larger caloric restriction and/or increased exercise. However, we think that the fundamental factor is that in the population herein reported, the percentage of prepregnancy overweight/obesity is much lower than in the aforementioned meta-analysis (34.2% vs. 68%) and we have also seen that this is associated to less eGWG. 

### 4.2. Variables Independently Associated with Inadequate GWG

The study has identified seven independent variables for inadequate GWG, namely non-Caucasian ethnicity, unfavorable obstetric history, maternal height, prepregnancy BMI category, smoking habit, gestational age at delivery and length of follow-up for GDM.

For iGWG, independent variables were unfavorable obstetric history, height, prepregnancy overweight /obesity, active smoking/quitting smoking during pregnancy, gestational age at delivery and length of follow-up, all of them negatively associated with the exception of length of follow-up. For eGWG, independent variables were non-Caucasian ethnicity, prepregnancy overweight /obesity, active smoking/quitting smoking during pregnancy; maternal height and length of follow-up displayed borderline significance.

#### 4.2.1. Ethnicity

The association of ethnicity with eGWG herein described is in line with information in the literature of inadequate GWG being different according to ethnicity both in the general obstetric population [32,33] and in women with GDM [31].

#### 4.2.2. Maternal Anthropometry

The observation that height is associated with GWG is in line with the report of Straube et al., describing that for a given maternal BMI, weight gain during pregnancy increased with maternal height [34], and also with the more recent report of Khanolkar et al., describing that maternal height increases with the category of GWG (insufficient < adequate < excessive) [35].

Our observation of a negative association between prepregnancy overweight/obesity and iGWG and a positive one with eGWG is broadly in line with current information both in the general pregnant population and in women with GDM. In the general pregnant population, Lindberg et al. described that underweight was associated with a higher risk of iGWG, and overweight, and obesity class I and II with eGWG [36]. In women with GDM, Wong et al. also have described that, in relation to women gaining within recommendations, those with eGWG had a higher prepregnancy BMI (28.4 vs. 25.0 kg/m^2^), whereas those gaining less than recommended displayed no differences [31].

#### 4.2.3. Unfavorable Obstetric History

We attribute the lower odds of insufficient GWG (OR 0.520, CI 95% 0.387–0.700) with unfavorable obstetric history to the fact that the latter was usually due to a macrosomic baby in a prior pregnancy and eGWG is a well-known risk factor for this condition [35]. We speculate that women with a prior macrosomic baby had eGWG in a prior pregnancy and repeated the pattern of GWG in current pregnancy.

#### 4.2.4. Smoking

It is well-known that nicotine increases energy expenditure, and may reduce appetite, so that smokers tend to have lower body weight and at the same time smoking cessation is commonly followed by weight gain both outside [37] and during pregnancy [36,38]. In the same line, in the current study women beginning pregnancy as smokers and quitting during pregnancy had a higher odds of eGWG (OR 1.614, 1.121–2.325) and a lower odds of iGWG (OR 0.668, 0.490–0.910). However, the observation that women who continued smoking during pregnancy also had a similar pattern of GWG can seem counterintuitive at first glance. Our interpretation is that women who continue to smoke during pregnancy reduce the consumption of cigarettes to a similar or higher extent than those who quit. In the last 10 years, where specific information on number of cigarettes consumed has been collected (N = 234), women who continued to smoke during pregnancy reduced the number of cigarettes in the same range than those who stopped (from 16.9 to 5.9 vs. 8 to 0, data not shown). Thus, the reduction in cigarettes per day would be similar in women who continued smoking during pregnancy and those who quitted, and similar patterns of GWG could be expected. Other studies in the general obstetric population have associated active smoking either with insufficient GWG [36] or smoking (unclear definition) with a twofold risk of excessive GWG [9]. 

#### 4.2.5. Gestational Age at Delivery and Length of Follow-up

We observed that earlier gestational age at delivery and longer follow-up was associated with a higher frequency of iGWG. The interpretation of higher odds of iGWG with shorter duration of pregnancy is straightforward. As to length follow-up, although we do not have specific information on weight gain before and after initiation of specific follow-up for GDM, we attribute overall results of GWG according to IOM to the impact of the intervention for GDM. The diet initially prescribed is normocaloric but it is modified afterwards (by healthcare providers and by pregnant women themselves) to achieve the metabolic goals [39]. This is in line with the observations of Berglund et al., where women with GDM had a lower total GWG versus women with normal glucose tolerance at the expense of a lower GWG after diagnosis [18], and with those of Hillier et al., who recently reported that obese women with GDM had less eGWG when diagnosed after early screening than at 24–28 weeks (35 vs. 59%) [40]. The current report establishes this fact in a much larger population.

As to independent predictors of GWG, only length of specific follow-up for GDM can be considered as modifiable during pregnancy, in contrast with smoking habit and gestational age at delivery. Follow-up for GDM is required for the treatment of the condition. In addition, taking into account the essentially satisfactory pregnancy outcomes with iGWG, poorer outcomes with eGWG and the association of length of follow-up with both (borderline with eGWG), we conclude that follow-up for GDM likely has an impact on outcomes through GWG.

### 4.3. Inadequate GWG and Pregnancy Outcomes

#### 4.3.1. PIH and Preeclampsia

With regard to pregnancy outcomes, in the adjusted analysis, hypertensive disorders (both PIH and preeclampsia) were associated with GWG categories even when, individually, iGWG and eGWG were not significantly different from aGWG. The direction of the association was towards a higher risk with eGWG and a lower risk with iGWG. The magnitude of the association was nominally larger for preeclampsia than for PIH (i.e., the aOR and 95%CI for eGWG was 6.519, 0.746–56.939 for preeclampsia vs. 1.357, 0.818–2.253 for PIH).

These observations are in line with data from previous investigations. In the general obstetric population Fortner et al. reported that women with eGWG had a ≃3-fold increased risk for hypertension and a ≃4-fold increased risk for preeclampsia, compared with aGWG [41]. On the other side, there is a reduced risk of hypertensive disorders in association with iGWG [42]. Among women with GDM, the abovementioned meta-analysis also observed a significant association with eGWG with hypertensive disorders of pregnancy (OR 1.65) but no association with iGWG [13]. Interestingly, GWG is positively associated with concurrent blood pressure in all gestational periods [41] and some authors have only observed an association in the third trimester (i.e., Gaillard for preeclampsia [9] or Gonzalez et al. for PIH [43]). Thus, GWG from the diagnosis of GDM onwards could play a relevant a role in the risk of PIH and preeclampsia and we have shown that the length of specific follow-up for GDM is negatively associated with GWG supporting a beneficial role of GDM care.

#### 4.3.2. Cesarean Delivery

We also have observed an association of GWG with cesarean delivery: the risk was higher in women with eGWG (aOR 1.641, 95 % CI 1.225–2.197) and lower in those with iGWG (0.715, 0.556–0.920). This is in agreement with information in the general obstetric population [44] and partially with data in women with GDM where the negative association of iGWG with cesarean delivery did not reach significance [13]. Since GWG in second [45] and third trimesters [46] has been related with cesarean delivery, we consider that the relationships described in this study are partially attributable to GWG taking place after GDM diagnosis and affected by its management.

#### 4.3.3. LGA, Macrosomia and SGA

In this series, eGWG was associated with a higher odds of LGA (aOR 2.003, 95% CI 1.397–2.871) and macrosomia (aOR 1.822,95% CI 1.152–2.881), results that are perfectly in line with those observed in meta-analyses both in the general obstetric population (respective ORs of 1.85 and 1.95) [47] and in women with GDM (respective RRs of 2.08 and 1.87) [13]. As to SGA newborns, eGWG was associated with less risk (aOR 0.515, 95% CI 0.310–0.855), and iGWG was not significantly different from the reference category (aOR 1.228, 95% CI 0.893–1.689). This is also in line with meta-analysis results in women with GDM where summary figures for individual categories of iGWG (RR 1.40) and eGWG (RR 0.57) were not significantly different from aGWG [13]. However, in the general obstetric population, individual IOM categories display similar risks in terms of SGA and are significantly different from reference (OR 1.53 for iGWG and 0.66 for eGWG) [47]. Thus, the lack of significance in women with GDM is likely attributable to insufficient statistical power. The relationship between GWG and birth- weight-related outcomes is present throughout pregnancy [9,48], so we also consider that observed associations of GWG with these outcomes are partly attributable to weight gain after GDM diagnosis.

We have not observed significant associations of GWG with the additional 12 outcomes addressed in the current investigation. Viecelli et al. did not show an association between GWG with either drug treatment or preterm birth. They did not address the other 10 variables [13]. Overall, GWG was not associated with neonatal hypocalcemia, but the category of iGWG displayed a positive association (OR 4.557, 95% CI 1.037–20.003). The association could be mediated through reduced birthweight [49] and requires confirmation.

### 4.4. Is iGWG Satisfactory in Women with GDM?

In relation to pregnancy outcomes, eGWG is associated with higher risks, and iGWG with a more favorable pattern. In fact, different authors have suggested that definition of adequate GWG should use more stringent limits in women with GDM [31,50], with approaches such as subtracting 2 kg to the limits indicated by IOM [31] or deriving specific limits using ROC curves [51].

iGWG during pregnancy also raises the issue of ketogenesis, known to be associated with neurocognitive outcomes in the offspring [52]. It is usually recommended to monitor ketones in women with GDM displaying overt hyperglycemia and/or weight loss during follow-up even when this has not been formally tested to improve fetal outcomes [53]. In fact, it has been our practice for decades [54] to recommend urinary ketone monitoring in women with GDM and not only before breakfast [55] in order to identify its presence and introduce modifications for its prevention. Thus, we consider that women in this series, have not displayed significant amounts of urinary ketones during follow-up for GDM or at least, not for long periods.

The observed GWG distribution can be viewed as a good outcome according to the observed association with pregnancy outcomes. However, a recent study in a general obstetric population in Hong Kong has reported that inadequate GWG is associated with adiposity, hypertension and insulin resistance in offspring at 7 years of age, independently of factors such as gestational hyperglycemia or birthweight [56]. The association is present both for excessive and insufficient GWG but more marked for eGWG and especially for extreme values. The relationship of iGWG with long-term adiposity and insulin resistance would be akin to situations such as maternal treatment with metformin [57,58] or prevention of GDM through lifestyle intervention [59]. Should these observations be confirmed in women with GDM, the distribution of GWG herein described should be viewed as less satisfactory.

The results herein presented confirm data in the literature and also include novel observations among predictors of GWG (maternal height, active smoking and length of follow-up) and also among its association with pregnancy outcomes (iGWG associated with a lower rate of cesarean delivery). 

The strength of our retrospective study is the large sample size of the cohort. We provide data on more than two thousand women with GDM and have performed a comprehensive analysis of the predictors of inadequate GWG according to IOM 2009 and its association with pregnancy outcomes. The study has several limitations. This is an observational, retrospective, single-center study, hence selection and information bias cannot be ruled out. A second limitation is that only information on total GWG is available and we have not been able to address weight gain after initiation of specific follow-up for GDM. Another limitation is that information on preeclampsia is only available in a reduced subset of women. 

## 5. Conclusions

In summary, in this cohort of women with GDM, inadequate GWG was very common at the expense of iGWG; length of clinical follow-up for GDM was the only independent variable that is modifiable during pregnancy, after GDM diagnosis. The associations of GWG with pregnancy outcomes were essentially favorable for iGWG and unfavorable for eGWG.

## Figures and Tables

**Figure 1 jcm-09-03343-f001:**
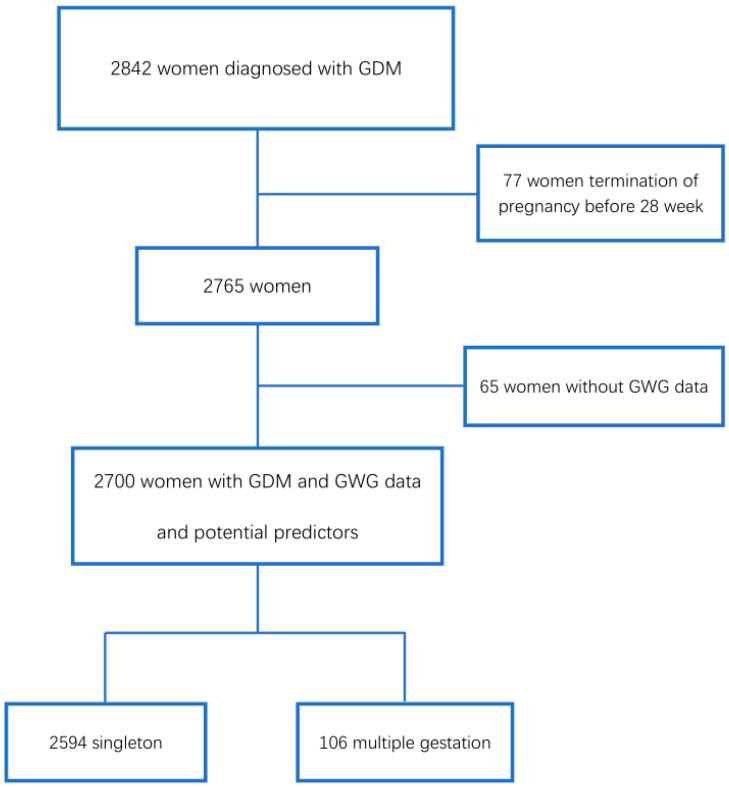
Flowchart of inclusion of patients in the study. GDM: Gestation Diabetes Mellitus. GWG: Gestational Weight Gain.

**Figure 2 jcm-09-03343-f002:**
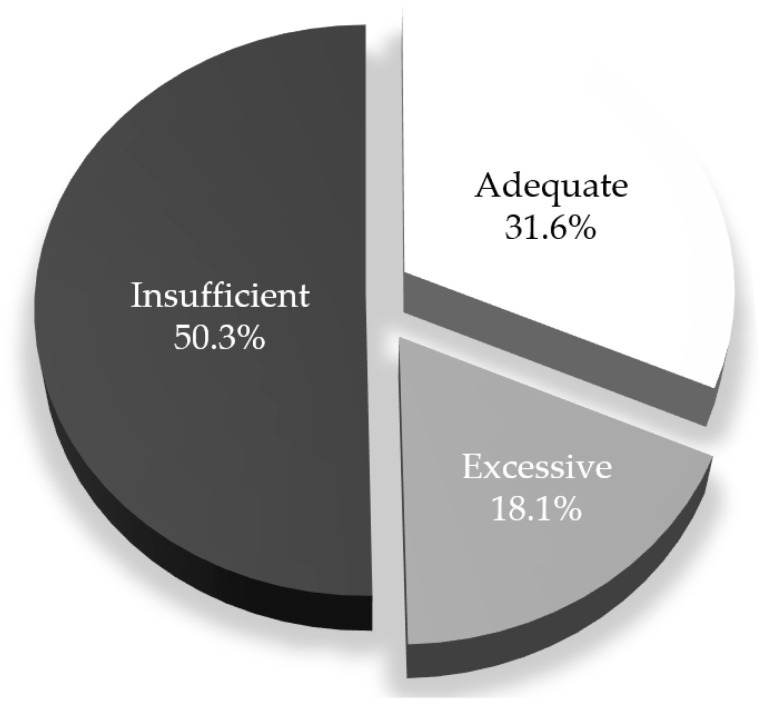
Distribution of gestational weight gain according to Institute of Medicine 2009.

**Table 1 jcm-09-03343-t001:** Characteristics of the study participants, both overall and according to category of weight gain after Institute of Medicine 2009.

Characteristic	% or P50 (P25, P75) in Each Weight Gain Category				*p* within IOM Categories *
	Overall	Weight Gain < IOM	Weight Gain within IOM	Weight Gain > IOM	
**Non-Caucasian ethnicity (%)**	5.5	4.4	3.8	11.3	<0.001
Age (years)	33.0 (29.0; 36.0)	33.0 (29.0; 36.0)	33.0 (30.0; 36.0)	33.0 (29.0; 36.0)	0.694
**Height (cm)**	160 (155; 164)	159 (155; 163)	160 (156; 164)	160 (156; 164)	0.002
**Prepregnancy BMI category (%)** Underweight Normal weight Overweight Obesity	2.6 63.1 23.5 10.7	2.7 77.9 12.3 7.1	3.4 58.5 28.8 9.3	1.0 30.1 45.5 23.4	<0.001
**Family history of diabetes (%)**	56.4	54.4	57.7	59.8	0.083
Prior history of abnormal glucose tolerance/gestational diabetes mellitus (%)	13.7	13.5	13.5	14.7	0.783
**Prior pregnancy (%)**	62.3	60.0	64.6	65.0	0.041
**Unfavorable obstetric history (%)**	12.8	9.3	15.3	18.4	<0.001
**Multiple pregnancy (%)**	3.9	5.7	2.3	1..6	<0.001
**Smoking habit during pregnancy** quitter (%) active smokers (%)	11.5 23.4	9.5 20.8	12.2 24.7	15.8 28.2	<0.001
Season at gestational diabetes mellitus diagnosis summer autumn winter	31.0 25.2 19.3	30.6 24.6 20.1	30.7 26.7 17.6	32.4 24.2 20.1	0.681
**Gestational age at diagnosis of gestational diabetes mellitus (weeks)**	29 (26; 33)	29 (25; 33)	30 (26; 33)	29 (26; 34)	0.008
Glycemic values (mmol/L) at diagnosis **Glucose 0 h** Glucose 1 h Glucose 2 h **Glucose 3 h**	4.7 (4.3; 5.1) 11.6 (10.9; 12.5) 10.2 (9.6; 11.1) 7.8 (6.6; 8.8)	4.6 (4.3; 5.0) 11.6 (10.9; 12.5) 10.2 (9.6; 11.1) 7.9 (6.8; 8.8)	4.7 (4.31; 5.1) 11.6 (10.8; 12.4) 10.2 (9.6; 11.1) 7.9 (6.6; 8.8)	4.9 (4.5; 5.4) 11.5 (10.9; 12.6) 10.2 (9.5; 11.1) 7.5 (6.1; 8.7)	<0.001 0.562 0.183 0.002
Number of abnormal glucose values	2 (2; 3)	2 (2; 3)	2 (2; 3)	2 (2; 3)	0.594
Autoimmunity against beta cells (%)	9.2	9.0	9.4	9.6	0.904
**Gestational age at delivery (weeks)**	39 (38;40)	39 (38;40)	39 (38;40)	39 (38;40)	<0.001
**Length of follow-up (weeks)**	7 (3; 11)	7 (4; 11)	7 (4;10)	6 (3;10)	0.032
Weight gain (kg)	10.2 (7.7; 13.0)	8.2 (5.7; 10.0)	12.2 (9.5; 13.9)	16.0 (13.0; 18.2)	<0.001

IOM: Institute of Medicine; BMI: body mass index; * Variables different from weight gain with a *p* value < 0.100 after bivariate multinomial regression analysis are displayed in bold characters and used in the multivariate multinomial logistic regression analysis.

**Table 2 jcm-09-03343-t002:** Independent variables for insufficient or excessive gestational weight gain according to IOM 2009 (multinomial multivariate logistic regression analysis).

	Insufficient Weight Gain				Excessive Weight Gain		
	OR	*p*	95% CI	Overall *p*	OR	*p*	95% CI
Non-Caucasian ethnicity (yes)	1.182	0.500	0.727–1.922	<0.001	**3.283**	**<0.001**	**1.984–5.433**
Height (cm)	**0.979**	**0.008**	**0.963–0.994**	0.001	1.020	0.062	0.999–1.041
Prepregnancy BMI category Underweight Normal weight (reference category) Overweight Obesity	0.659 1 **0.301** **0.603**	0.137 **<0.001** **0.005**	0.381–1.142 **0.235–0.387** **0.425–0.857**	<0.001	0.487 1 **3.190** **5.060**	0.191 **< 0.001** **< 0.001**	0.165–1.433 **2.403–4.234** **3.472–7.375**
Unfavorable obstetric history (yes)	**0.520**	**<0.001**	**0.387–0.700**	<0.001	0.865	0.399	0.618–1.211
Smoking habit Non-smoker (reference category) Quitter during pregnancy (yes) Active smoker during pregnancy (yes)	1 **0.668** **0.727**	**0.011** **0.007**	**0.490–0.910** **0.576–0.918**	<0.001	1 **1.614** **1.382**	**0.010** **0.030**	**1.121–2.325** **1.031–1.852**
Gestational age at delivery (weeks)	**0.855**	**<0.001**	**0.806–0.907**	<0.001	1.026	0.527	0.947–1.113
Length of follow-up (weeks)	**1.035**	**<0.001**	**1.018–1.052**	<0.001	0.980	0.066	0.958–1.001

OR: odds ratio; CI: confidence interval; *p* value < 0.05 are considered significant; ORs significantly different from the reference category are marked in bold.

**Table 3 jcm-09-03343-t003:** Pregnancy outcomes of women with gestational diabetes mellitus according to gestational weight gain category.

Outcome	Prevalence (%) in Each GWG Category				Overall *p*
	Overall	iGWG	aGWG	eGWG	
Pregnancy-induced hypertension	5.3	3.4	5.5	10.1	<0.001
Preeclampsia	2.9	1.3	1.7	6.5	0.031
Insulin treatment	46.8	45.4	46.5	51.4	0.068
Cesarean delivery	24.1	19.9	23.4	36.8	<0.001
Fetal scalp blood pH < 7.25	3.8	3.1	4.0	5.3	0.077
Preterm birth	9.8	13.3	6.9	4.8	<0.001
Apgar at 5 min < 7	0.5	0.6	0.3	0.4	0.735
Arterial pH < 7.1	3.8	3.5	3.1	6.0	0.035
Obstetric trauma	2.3	2.1	2.6	2.4	0.697
LGA newborn	11.2	6.4	10.7	26.1	<0.001
Macrosomia (≥ 4000 g)	5.7	2.4	5.7	14.9	<0.001
SGA newborn	9.9	11.7	9.3	5.9	0.001
Neonatal hypoglycemia	2.5	2.4	2.5	2.7	0.932
Jaundice requiring treatment	5.1	6.1	4.0	4.1	0.040
Respiratory distress requiring treatment	3.3	4.7	1.9	2.0	<0.001
Neonatal hypocalcemia	1.6	2.5	1.0	0.0	0.009
Neonatal polycythemia	1.4	1.3	1.4	1.8	0.662
Perinatal mortality	0.5	0.6	0.5	0.2	0.603

GWG (gestational weight gain), iGWG (insufficient gestational weight gain), aGWG (adequate gestational weight gain), eGWG (excessive gestational weight gain), LGA (large-for-gestational age), SGA (small-for-gestational age). A Chi-square test was used for statistical analyses.

**Table 4 jcm-09-03343-t004:** Risk of different pregnancy outcomes in women with gestational diabetes mellitus according to gestational weight gain.

Outcome	Unadjusted OR Unadjusted CI 95%				Adjusted OR * Adjusted CI 95%			
	iGWG	aGWG	eGWG	Overall *p*	iGWG	aGWG	eGWG	Overall *p*
Pregnancy-induced hypertension	**0.604** **0.397–0.920**	**1** **1**	**1.949** **1.282–2.963**	<0.001	0.655 0.396–1.085	1 1	1.357 0.818–2.253	0.028
Preeclampsia	0.785 0.109–5.658	1 1	4.095 0.832–20.161	0.054	1.008 0.077–13.163	1 1	6.519 0.746–56.939	<0.050
Insulin treatment	0.955 0.805–1.134	1 1	1.219 0.976–1.523	0.068	--	1 1	--	ns
Cesarean delivery	**0.812** **0.660–0.999**	**1** **1**	**1.898** **1.488–2.419**	<0.001	**0.715** **0.556–0.920**	**1** **1**	**1.641** **1.225–2.197**	<0.001
Fetal scalp blood pH < 7.25	0.777 0.492–1.225	1 1	1.367 0.810–2.306	0.081	--	1 1	--	ns
Preterm birth	**2.029** **1.546–2.832**	1 1	0.689 0.424–1.121	<0.001	--	1 1	--	ns
Apgar at 5 min < 7	1.647 0.436–6.226	1 1	1.182 0.197–7.097	0.739	--	1 1	--	ns
Arterial pH < 7.1	1.138 0.672–1.927	1 1	**2.000** **1.113–3.595**	0.038	--	1 1	--	ns
Obstetric trauma	0.793 0.453–1.389	1 1	0.949 0.466–1.936	0.698	--	1 1	--	ns
LGA newborn	**0.575** **0.425–0.777**	**1** **1**	**2.952** **2.200–3.962**	<0.001	**0.569** **0.400–0.810**	1 1	**2.003** **1.397–2.871**	<0.001
Macrosomia (≥4000g)	**0.412** **0.265–0.640**	**1** **1**	**2.893** **1.983–4.220**	<0.001	**0.461** **0.282–0.752**	1 1	**1.822** **1.152–2.881**	<0.001
SGA newborn	1.289 0.947–1.706	1 1	**0.608** **0.392–0.943**	0.001	1.228 0.893–1.689	1 1	**0.515** **0.310–0.855**	0.002
Neonatal hypoglycemia	0.951 0.547–1.656	1 1	1.076 0.534–2.168	0.932	--	1 1	--	ns
Jaundice requiring treatment	**1.579** **1.052–2.369**	1 1	1.024 0.583–1.800	0.041	--	1 1	--	ns
Respiratory distress requiring treatment	**2.558** **1.472–4.447**	1 1	1.086 0.489–2.413	0.001	--	1 1	--	ns
Neonatal hypocalcemia	2.529 0.952–6.720	1 1	odd coefficient	0.177	**4.557** **1.037–20.003**	1 1	odd coefficient	0.133
Neonatal polycythemia	0.904 0.433–1.886	1 1	1.310 0.548–3.132	0.665	--	1 1	--	ns
Perinatal mortality	1.213 0.364–4.039	1 1	0.437 0.049–3.917	0.624	--	1 1	--	ns

OR (odds ratio), CI (confidence interval), iGWG (insufficient gestational weight gain), aGWG (adequate gestational weight gain), eGWG (excessive gestational weight gain), LGA (large-for-gestational age), SGA (small-for-gestational age). Logistic regression analysis was used to calculate ORs. * See methods for variables used for adjustment. ORs significantly different from the reference category are marked in bold, —is indicated when IOM is not included in the last step and OR not available.

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
