# Peer review of "Inadequate Weight Gain According to the Institute of Medicine 2009 Guidelines in Women with Gestational Diabetes: Frequency, Clinical Predictors, and the Association with Pregnancy Outcomes"

_jcm, 2020, doi:10.3390/jcm9103343_

Round 1

Reviewer 1 Report

In this study, Xie et al. evaluated the rate of inadequate gestational weight gain (GWG) in women with GDM and associated independent variables. This was a retrospective study including 2842 pregnant women with GDM between 1985 and 2011. Independent variables for insufficient GWG were height, pre-pregnancy overweight/obesity, smoking habit and length of GDM follow-up.

The relatively large sample size is a strength of the study. There are some comments.

  1. It was not clear what the marks (-) or (+) meant. This should be explained clearly.
  2. The authors concluded that “inadequate GWG was very common”, but the data was not presented. The data should be described in Results.
  3. Definition of IOM 2009 for GWG should be explained more clearly in the manuscript.
  4. The authors evaluated the “adjusted” GWG, the GWG linearly adjusted to a gestational age of 40 weeks. This adjusted GWG might induce the error since GWG usually does not increase linearly during the gestation. Having this large sample size, the authors are also recommended to analyze using the “actual” GWG during pregnancy in the women with labor at term.
  5. The patients were enrolled between 1985 and 2011. During this period, diagnosis criteria and management of GDM might be changed. This point should be taken into account for the analyses and discussed in the manuscript.
  6. The association between the length of GDM follow-up and insufficient GWG appears the effect of GDM treatment. Nutritional therapy for GDM should be more clearly described in the Methods section.
  7. Clinical consequence of insufficient GWG was not clear. The association between insufficient GWG and perinatal outcomes should be assessed.
  8. Despite the relatively large sample size, the results were mostly confirmative. The novelty of study should be more clearly stated.

Author Response

Point 1: It was not clear what the marks (-) or (+) meant. This should be explained clearly.

Response 1: These expressions are no longer present in the abstract (see lines 35-39 of tracked manuscript).

Point 2: The authors concluded that “inadequate GWG was very common”, but the data was not presented. The data should be described in Results.

Response 2: We thank the reviewers for this indication; the rates of GWG are now shown in lines 33-34 of the tracked document.

Point 3: Definition of IOM 2009 for GWG should be explained more clearly in the manuscript.

Response 3: In the current version of the manuscript, we have added Appendix A including this information (lines 449-452 of the tracked document).

Point 4: The authors evaluated the “adjusted” GWG, the GWG linearly adjusted to a gestational age of 40 weeks. This adjusted GWG might induce the error since GWG usually does not increase linearly during the gestation. Having this large sample size, the authors are also recommended to analyze using the “actual” GWG during pregnancy in the women with labor at term.

Response 4: Gestational age at delivery has an influence on GWG but we agree that a linear adjustment to 40 weeks could induce an error. We have now performed the analysis using actual GWG (definition in lines 88-91 of the tracked document).

Point 5: The patients were enrolled between 1985 and 2011. During this period, diagnosis criteria and management of GDM might be changed. This point should be taken into account for the analyses and discussed in the manuscript.

Response 5: We have now indicated changes throughout the years in diagnosis, treatment (lines 81-86 of the tracked document) and outcome definition (lines 112-115 and 125-129 of the tracked document).

Point 6: The association between the length of GDM follow-up and insufficient GWG appears the effect of GDM treatment. Nutritional therapy for GDM should be more clearly described in the Methods section.

Response 6: We have indicated in the methods section that nutrition aims at a normocaloric diet (lines 83-86 of the tracked manuscript) and in the discussion that it is modified by both women and healthcare professionals to achieve metabolic goals (lines 242-247) but also taking ketonuria into account (lines 403-409).

Point 7: Clinical consequence of insufficient GWG was not clear. The association between insufficient GWG and perinatal outcomes should be assessed.

Response 7:  In the current version of the study we have included the crude and adjusted association of GWG with major maternal and fetal outcomes (lines 103-129 and 143 -147 in the methods section; lines 189-224 in the results section and lines 347-418 in the discussion section).

Point 8: Despite the relatively large sample size, the results were mostly confirmative. The novelty of study should be more clearly stated.

Response 8: We thank the reviewer for this suggestion. We have included a novelty statement in lines 419-422 of the tracked revised manuscript.

Reviewer 2 Report

Weight gain in women with gestational diabetes 2 mellitus. Clinical predictors of inadequate weight 3 gain according to IOM 2009

Summary: Current manuscript summarizes the results from the retrospective analysis of data collected over the period of ~25 years on pregnant patients with Gestational diabetes Mellitus (GDM) at the Endocrinology and Pregnancy clinic at Hospital De La Santa Creu i Sant pau. The authors conclude that inadequate Gestational weight gain was very common in the current cohort and the length of the follow up for GDM is the only modifiable independent variable during pregnancy.

Comments:

  1. Can the authors state what is the major finding from this study that is different from that of the earlier studies (Viecceli et al Obesity Reviews, 2017 and Wong et al Diabetologia, 2017)? It does not seem to be highlighted in this current manuscript.
  2. For some of the variables that were analyzed in the study, it will be useful and easier to look as a correlation graph, instead of just table. Such as inadequate weight gain vs height etc.
  3. In the Table 1, the authors should explain how the percentages are calculated? It is slightly confusing in the current format. Best would be to describe it underneath the table.
  4. The title of the manuscript is too long and ambiguous.

Overall, the manuscript seems well organized and concise. However, the authors need to highlight or bring out the novel findings form this study that are unique to the current study.

Author Response

Point 1: Can the authors state what is the major finding from this study that is different from that of the earlier studies (Viecceli et al Obesity Reviews, 2017 and Wong et al Diabetologia, 2017)? It does not seem to be highlighted in this current manuscript.

Response 1: We thank the reviewer for this suggestion. We have included a novelty statement in lines 419-422 of the tracked revised manuscript.

Point 2: For some of the variables that were analyzed in the study, it will be useful and easier to look as a correlation graph, instead of just table. Such as inadequate weight gain vs height etc.

Response 2: We agree with the reviewer that associations are easier to grasp using a graphical display. We have included the scatterplot of GWG vs length of follow-up in Appendix B (lines 454-456 of the revised manuscript). We have not included more variables due to time constraints and because the real outcome (GWG according to IOM) was not amenable to a scatterplot display.

Point 3: In the Table 1, the authors should explain how the percentages are calculated? It is slightly confusing in the current format. Best would be to describe it underneath the table.

Response 3: We thank the reviewer for pointing to a section of the document that is not clear. We have now indicated that % are calculated within each IOM category (lie 201 of the tracked document).

Point 4: The title of the manuscript is too long and ambiguous.

Response 4: We have reworded the title, we expect it is now more clear (lines 2-6 of the tracked document).

Round 2

Reviewer 1 Report

The authors have responded to the comments appropriately and the manuscript has been improved.

Author Response

We want to thank the reviewer for his/her comments